# Extreme Low Cytosolic pH Is a Signal for Cell Survival in Acid Stressed Yeast

**DOI:** 10.3390/genes11060656

**Published:** 2020-06-16

**Authors:** Rodrigo Mendonça Lucena, Laura Dolz-Edo, Stanley Brul, Marcos Antonio de Morais, Gertien Smits

**Affiliations:** 1Department of Genetics, Biosciences Centre, Federal University of Pernambuco, Recife 50670-901, Brazil; lucenarm2@gmail.com; 2Molecular Biology and Microbial Food Safety, Swammerdam Institute for Life Sciences, University of Amsterdam, 1098 XH Amsterdam, Netherlands; ldolzedo@gmail.com (L.D.-E.); s.brul@uva.nl (S.B.)

**Keywords:** calcium signalling, cell wall integrity pathway, cytosolic pH, membrane potential, PKC signalling, sulphuric acid

## Abstract

Yeast biomass is recycled in the process of bioethanol production using treatment with dilute sulphuric acid to control the bacterial population. This treatment can lead to loss of cell viability, with consequences on the fermentation yield. Thus, the aim of this study was to define the functional cellular responses to inorganic acid stress. *Saccharomyces cerevisiae* strains with mutation in several signalling pathways, as well as cells expressing pH-sensitive GFP derivative ratiometric pHluorin, were tested for cell survival and cytosolic pH (pH_c_) variation during exposure to low external pH (pH_ex_). Mutants in calcium signalling and proton extrusion were transiently sensitive to low pH_ex_, while the CWI *slt2*Δ mutant lost viability. Rescue of this mutant was observed when cells were exposed to extreme low pH_ex_ or glucose starvation and was dependent on the induced reduction of pH_c_. Therefore, a lowered pH_c_ leads to a complete growth arrest, which protects the cells from lethal stress and keeps cells alive. Cytosolic pH is thus a signal that directs the growth stress-tolerance trade-off in yeast. A regulatory model was proposed to explain this mechanism, indicating the impairment of glucan synthesis as the primary cause of low pH_ex_ sensitivity.

## 1. Introduction

Yeast growing cells can face extracellular pH (pH_ex_) changes when they naturally acidify the environment because of the action of metabolism, or during human usage in biotechnological processes. Human-made acidic environments occur, for instance, in the process of bioethanol production in which yeast biomass is recycled using sulphuric acid to control the bacterial population [1,2], by addition of weak organic acids (WOAs) to prevent food contaminations [3], or even for probiotic yeast which has to pass through the acidic gastrointestinal tract [4]. The nature of the plasma membrane architecture makes it almost impermeable to protons, which keeps cytosolic pH (pH_c_) highly regulated. However, changes in environmental conditions can lead to changes in pH_c_, which is a highly dynamic property of cells [5,6]. The proton gradient over the plasma membrane provides the energy for the import of many nutrients, including amino acids, phosphate, several (non-preferred) sugars and other carbon sources [7]. Uptake of these nutrients, through proton symport, results in the influx of protons. The cells counteract this acidification through the activation of the plasma membrane H^+^-ATPase pump Pma1p, which exports the protons at the cost of ATP [8]. The activity of Pma1p is thought to consume around 15% of the ATP produced in metabolically active yeast.

Besides these metabolic effects on intracellular pH, pH_c_ can also actively respond to environmental changes, such as temperature, osmolarity and, importantly, the abundance of a readily fermentable carbon source such as glucose [5,9,10]. Increases in pH_c_ are correlated with cell proliferation [11,12], differentiation [13], cell cycle progression [14] and apoptosis [15]. In addition, development of cancers cells depends on both extracellular pH (pH_ex_) and pH_c_ [16,17]. With pH_ex_ equal or below the pKa, WOAs such as acetic or sorbic acid prevail in undissociated form in which they can pass through plasma membrane. In the neutral cytosol, the proton dissociates from the acid, leading to the accumulation of protons and anions in the cell interior [6,18]. The acidification is countered by the activity of the already expressed Pma1p, helped by the induced expression of the ABC transporter Pdr12p to pump out the sorbate and some moderately lipophilic carboxylic anions [18]. However, this transporter does not act against dicarboxylic and highly lipophilic long-chain fatty acids [19].

In addition, yeast responds to low pH_ex_ by remodelling its cell wall [20]. Weak Organic Acids (WOAs) cause a decrease of growth because the reduction of the pH_c_ and the energy spent to extrude anions and protons from the cell [12,21]. For strong inorganic acids such as hydrochloric or sulphuric acids, adaptation of yeast to low pH involves mechanisms that include induction of Cell Wall Integrity (CWI) genes [2,22] and General Stress Response (GSR) pathway [1] in which the action of the protein kinase C (PKC) pathway is essential [23,24,25]. Transient inactivation of glucose sensing pathway and reduction of growth have been reported and suggest a reduction of pKA activity, which in turn releases the GSR, leading the cells to reprogram their gene expression to adapt to the low pH environment [1,2]. Additionally, the response to low pH_ex_ is dependent on calcium metabolism [26], and deletion of calcium channels Mid1p or Cch1p is lethal to the yeast cells under inorganic acid stress [23,25]. There is a coordinated action between PKC pathway and High Osmotic Glycerol (HOG) response indicating that damages occurring at the cell wall structure also impose some sort of osmotic stress to the yeast cells [25,27].

Despite the evidence pointing to damage in the cell wall during exposure of the yeast to inorganic acids, there is little understanding of the effects of the low environmental pH in the cytosolic pH and their consequences. In the present work, we went further in the investigation of the mode of action of low pH in yeast cells, asking whether low pH_ex_ exerts intracellular effects. We showed that, while survival at low pH_ex_ requires proper cell wall maintenance or repair, a reduction of pH_c_ leading to reduced growth can compensate for the lack of this CWI mediated process.

## 2. Materials and Methods

### 2.1. Strains and Growth Conditions

*S. cerevisiae* BY4741 (MATα *his3*Δ *leu2*Δ *met15*Δ *ura3*Δ) strain and its isogenic deletion mutants from the EUROSCARF collection were used (Table 1). The cells were grown in Yeast Peptone Dextrose (YPD) medium (10 g/L yeast extract, 20 g/L peptone and 20 g/L dextrose) or Yeast Nitrogen Base (YNB) medium (6.7 g/L YNB without amino acid, 2 g/L drop-out amino acid supplement powder mixture and 20 g/L glucose). Drop-out mixture was minus uracil Kaiser drop-out mix supplement and was purchased from ForMedium (Hunstanton, United Kingdom). Other chemicals were purchased from Sigma-Aldrich (St. Louis, MA, USA). Solid medium contained 20 g/L agar. To perform intracellular pH measurement, we used a low-fluorescence (LOFLO) synthetic defined monosodium glutamate medium. This medium contained 1.7 g/L YNB w/o ammonium sulphate w/o folic acid, w/o riboflavin (MP Biochemicals, Solon, OH, USA), 1 g/L glutamic acid sodium salt hydrate (Sigma-Aldrich), 2 g/L drop-out amino acid supplement powder mixture without uracil and 20 g/L glucose. Liquid pre-cultures in 10 mL glass tubes were incubated at 30 °C in orbital shaker. For cultivation in liquid medium, pH values were adjusted with concentrated sulphuric acid to various pH values as indicated and media were autoclaved or filter sterilised using 0.22 µm disposable sterile filters. For solid media, to allow agar solidification at low pH, we mixed the reagents in demineralised water and adjusted pH with sulphuric acid. The agar-containing media were sterilised by heating in a microwave oven for about 2 min in intervals of 15 s, with intermittent mixing, until the components were completely dissolved. Sterility test was performed by incubating the plates as a negative control in the experiments.

### 2.2. Growth and Recovery Determination

For pre-inocula, yeast cells from plates were grown overnight in flasks containing liquid YNB medium at 30 °C under orbital agitation. Cultures were diluted 10× and transferred to CELLSTAR black polystyrene clear bottom 96-well microtiter plates (BMG Labtechnologies, Ortenberg, Germany) containing LOFLO media at indicated pH. Microtiter plates were incubated at 30 °C with orbital shaking under orbital agitation at 200 rpm in a FLUOstar Optima microtiter plate reader (BMG Labtechnologies, Ortenberg, Germany). OD_600_ and fluorescence emissions for pH_c_ determination (see details in 2.5) of the cultures were measured over time in three technical replicates. To measure growth recovery, the cultures grown under different pH conditions in microtiter plates as explained above were collected by centrifugation and resuspended in fresh LOFLO medium with pH 5.0. After resuspension, cells were transferred to microtiter plates and incubated at 30 °C in a FLUOstar Optima microtiter as before, with measurement of both OD600 and fluorescence emissions. Experiments were performed in at least two biological replicates.

To correct OD600 for linearity, cultures were grown overnight to maximum cell density. Serial dilutions were made in three technical replicates in a microtiter plate and OD600 were measured in microtiter plate reader, with blank value subtracted. We plotted measured (x) vs. dilution corrected (y) OD600 and fitted to a second-order polynomial function to determine corrected ODs in all time-course experiments.

### 2.3. Spot Assays

Selected strains were grown overnight in liquid YPD or YNB medium. Culture concentrations were adjusted to OD600 = 1.0 using a UV/Vis spectrophotometer (Lightwave II Isogen Life Sciences, De Meern, The Netherlands). Four serial dilutions of 1/10 were suspended in saline solution in sterile round-bottom 96-well microplates. For each dilution, 4 µL were spotted on YPD or YNB plates adjusted to specific pH using sulphuric acid or supplemented with additives as described for testing (see figure legends). Plates were incubated at 30 °C for 2 days and photographed.

### 2.4. Cell Viability

To test the viability yeast cells, overnight pre-cultures were centrifuged and resuspended at OD600 2.0 in flasks with fresh LOFLO media adjusted to desired final pH with sulphuric acid. These cultures were incubated at 30 °C in an orbital shaker (200 rpm). Cultures were sampled every 2 h, and, after appropriate dilution in saline solution, 25 µL of cell suspension were spread on YNB plates with pH 5.0. These plates were incubated at 30 °C for 2 days. The number of colonies was recorded and used to calculate colony forming units (CFU). The experiments were performed using two biological and three technical replicates.

### 2.5. Cytosolic pH (pH_c_) Measurements 

Strains were transformed with plasmid pORY001 carrying the pH sensitive GFP variant ratiometric pHluorin under the control of the constitutively active actin promoter [5]. Transformation of yeast was performed by lithium acetate/single-stranded carrier DNA/polyethylene glycol method according [28]. pHluorin displays a bimodal excitation spectrum with peaks at 390 nm (protonated form) and 470 nm (unprotonated form) and an emission maximum at 510 nm. pHluorin calibration was performed as reported [5]. Briefly, a pH_c_ calibration curve was generated using cells mildly permeabilised with digitonin and incubated in buffers at pH 5.0–9.0. The ratios of emission intensity resulting from excitation at 390 and 470 nm, each emission value corrected for emission intensities of cultures expressing an empty vector, were plotted against calibration buffer pH. For live-cell pH determination, overnight pre-cultures were diluted 1/10 into CELLSTAR black polystyrene clear bottom 96-well microtiter plates (BMG Labtechnologies, Ortenberg, Germany) containing LOFLO media at indicated pH. Microtiter plates were incubated at 30 °C and orbital shaking in a FLUOstar Optima microtiter plate reader (BMG Labtechnologies, Ortenberg, Germany), and OD600 and fluorescence emissions of the cultures were measured in time in three technical replicates. Each experiment was performed in at least two biological replicates.

### 2.6. Protein Analysis 

The strains were grown overnight in liquid YNB, and then were transferred to sterile Erlenmeyer flasks (200 mL) with 60 mL of YNB media adjusted to pH 5.0 and 2.5 with sulphuric acid. Samples of 25 mL were collected directly upon inoculation (0 h) and after 8h of incubation at 200 rpm and 30 °C. Samples were transferred to 50 mL conical centrifuge tubes and centrifuged for 5 min to 5000 rpm. Twenty millilitres of the supernatant were filtered in a Corning^®^ syringe filters (0.2 µm) and transferred to a protein concentrator Corning^®^ Spin-X^®^ UF 20 (Sigma-Aldrich, St. Louis, MA, USA) which has a filter to protein cut off of 10,000 MW. Concentrators were placed in a swing bucket rotor and centrifuged to 5000× g to 4 °C for 20 min. After that, we recovered a volume of approximately 100 µL for each sample which were stored in microtubes in at −20 °C until mass spectrometer analysis.

For mass spectrometry, supernatant samples of the different yeast cultures were digested following an in-house protocol. Alkylation was done with iodoacetamide and 2 µg of trypsin was used for overnight digestion. The digestion was stopped with acidification by TFA and the peptides were collected using an 80 µg capacity OMIX RP tip (Agilent Technologies, Santa Clara, CA, USA). Eluted peptides had a final volume of 20 µL 50% ACN, 0.1% TFA and about 3–6 µL were dried in a speed-vac and reconstituted in 6 µL 2% ACN, 0.1% TFA for the analysis with LC-MS. An amaZon Speed Iontrap with a CaptiveSpray ion source (Bruker, Billerica, MA, USA) coupled to an EASY-nLC II (Proxeon, Thermo Scientific, Waltham, MA, USA) chromatographic system was used for the analysis. Samples (5 µL injection volume) were injected and separated with an eluent flow of 300 nL min^−1^ on an Acclaim PepMap100 (C18 75 µM 25 cm Dionex, Thermo Scientific, Waltham, MA, USA) analytical column combined with an Acclaim PepMap100 pre-column (C18 100 µM 2 cm Dionex, Thermo Scientific, Waltham, MA, USA) using a 30 min gradient of 0–50% ACN and 0.1% formic acid. Peptide precursor ions above a predefined threshold ion count were selected for low energy collision-induced dissociation (CID) to obtain fragmentation spectra of the peptides. After processing raw MS/MS data with Data Analysis software (Bruker), resulting mgf datafiles were used for database searching with licensed Mascot software v2.5.1 (Matrix Science, Boston, MA, USA) in a yeast database (Yeast strain ATCC 204508, UniProt). Searches were simultaneously performed against a “common contaminants database” (compiled by Max Planck Institute of Biochemistry, Martinsried) to minimise false identifications. A fixed modification of carbamidomethyl for cysteine, variable modification of oxidised methionine, and semitrypsin (due to the origin of the samples, possible proteolytic activity) with the allowance of one missed cleavage and peptide charge state +2, +3 and +4. Peptide and MS/MS mass error tolerances were 0.3 Da for ESI-trap. A Mascot minimal fidelity score of 30 was imposed. Duplicates and non-yeast proteins were removed from the list. Only proteins of which at least one peptide was detected in each biological sample were considered valid. The number of peptides identified was averaged over two biological replicates.

## 3. Results

### 3.1. Components of the PKC, Ca^2+^ Signalling Pathway and Cell Wall Biosynthesis Are Required for Low-pH Stress Tolerance 

We previously reported the importance of the PKC and Ca^2+^ signalling pathways in inorganic acid tolerance in rich (YPD) media [25]. Here, we expand these observations to establish the mode of action of low pH stress and how low pH exerts internal or external effects. As the phospholipid bilayer is almost impermeable to protons, low pH_ex_ by itself does not lead to a decrease of pH_c_ in minimal defined media [12] and does not cause growth inhibition under these conditions until at a pH_ex_ of 2.0 [25]. Hence, we chose pH_ex_ of 2.5 as the so-called Minimal Inhibitory pH (MIP), the condition that causes severe acid stress while still allowing growth of WT cells [25]. In YPD media, however, low pH_ex_ does affect WT growth, in all likelihood because it contains weak organic acid compounds that in a protonated state at low pH can pass the membrane [5]. To distinguish the direct effects of low pHex from indirect effects via weak organic acid compounds in the broth, we compared growth at low pHex in YPD and defined (YNB) media. WT growth is slightly reduced at pH_ex_ 2.5 compared to pH_ex_ 5.0 in both media (Figure 1, first row). As a control, we used *pma1-007*, a mutant with only 50% of the expression and activity of the proton pumping plasma membrane ATPase Pma1p [29,30], which cannot maintain proper pH_c_ at low pH_ex_ [12]. As expected, this mutant was hypersensitive to low pH_ex_ in both media plates (Figure 1, last row). After confirming the reliability of the spot test, we surveyed a large number of viable deletion mutants in the CWI pathway, as well as those of cell wall biosynthetic and cell wall protein encoding genes, as targets of the CWI pathway and their homologues (Figure 1). In the PKC signal transduction pathway, the results show hypersensitivity to pH_ex_ 2.5 of *wsc1*Δ mutant (among the mutants for sensor-encoding genes tested) and the *bck1*Δ and *slt2*Δ mutants (among the mutants for MAPK-encoding genes tested) in both media (Figure 1). Downstream of the kinase cascade, deletion mutants for the transcription factor-encoding genes *RLM1, SWI4* and *SWI6* were affected by low pH_ex_ in YPD, although they could grow in YNB at pH_ex_ 2.5 (Figure 1). Regarding CWI signalling and cell wall biogenesis mechanism, we observed the following picture: (a) *gas1*Δ mutant inactivated for β-1,3-glucanosyltransferase did not grow at pH_ex_ 2.5 in both media; (b) *chs1*Δ mutant inactivated for chitin synthase 1 was hypersensitive in YPD and mildly sensitive in YNB; and (c) *fks1*Δ mutant inactivated for the catalytic subunit of the 1,3-β-D-glucan synthase was hypersensitive in YPD only (Figure 1). All this confirmed that low pH_ex_ can be sensed at the cell envelope and requires PKC pathway mediated adaptation for normal growth, as previously indicated [25].

Ca^2+^ signalling is required for yeast to adapt to a variety of environmental stresses including low pH_ex_ stress [23,25,26]. Hence, we performed an analysis of deletion mutants for Ca^2+^ transport and signalling (Figure 2A). Hypersensitivity to low pH_ex_ in both media was observed for mutant cells carrying deletion of the *MID1* or *CCH1* genes, both encoding the subunits of a stretch-activated Ca^2+^-permeable cation channel and *PMR1* gene that codes for a high affinity Ca^2+^/Mn^2+^ P-type ATPase required for Ca^2+^ transport into the Golgi (Figure 2A). Inactivation of *ECM7* gene, coding for a protein involved in high affinity Ca^2+^ influx, also led to low pH_ex_ hypersensitivity in YPD, although mild sensitivity in YNB (Figure 2A). The mutants for the calcineurin, the Ca^2+^/calmodulin-regulated protein phosphatase, were also tested. The *cna2*Δ mutant lacking the catalytic subunit of calcineurin showed pH-sensitivity to growth in YPD but not in YNB media. However, the mutant for the regulatory subunit of calcineurin *cnb1*Δ was hypersensitive to growth at low pH_ex_ in both media (Figure 2A). In addition, the relevance of calcineurin to the yeast growth at low pH_ex_ was highlighted when we included the calcineurin inhibitor FK506 in the medium, which led to strong reduction of growth in WT at pH_ex_ 2.5 (Figure 2B). Together, these results show that growth at low pH_ex_ requires the influx of Ca^2+^ for activation of calcineurin. However, this need for calcium did not depend on the transcription factor Crz1p, because the *crz1*Δ mutant displayed similar phenotypes to WT at pH_ex_ 2.5 (Figure 2A). To test the dependence of low pH_ex_ sensitivity on Ca^2+^, we attempted a reduction and an increase of Ca^2+^ in the medium. Since Ca^2+^ chelation by EGTA depends on the negative charge of the chelators, protonation of the Ca^2+^ binding site at low pH_ex_ prevented chelation. In the inverse experiment, addition of extra Ca^2+^ to the medium did exert a strong effect and markedly improved growth at pH_ex_ 2.5 in wild type yeast (Figure 2C). A high concentration of Ca^2+^ in the media also recued the growth of the *mid1*Δ and *cch1*Δ mutants at pH_ex_ 2.5, but did not affect the *slt2*Δ mutant (Figure 2C). Hence, the Ca^2+^ in excess in the medium could be pumped-in at pH_ex_ 2.5 mediated by the low affinity Ca^2+^ influx system or by Ecm7 protein, in line with previous results [31], restoring the internal level that is required to cell survival and growth at low pH. However, this action seemed insufficient when cell wall integrity cannot be maintained.

### 3.2. Low pH_ex_ and High Osmolarity Exert Synergistic Stresses

To understand why CWI mutants are hypersensitive to external changes in proton concentration, we postulated that low pH_ex_ would damage the cell wall, requiring an adaptive response through the CWI pathway [25]. Cell wall damage leading to cell death can often be alleviated by osmotic stabilisation of the medium [32]. Illustrating this, the *slt2*Δ mutant was hypersensitive to the cell wall perturbing drug Congo red added to plates, but this could be compensated for by the addition of sorbitol (Figure 3A). In the case of low pH_ex_, however, sorbitol addition did not protect the cells, but rather enhanced the sensitivity (Figure 3B). Indeed, the combination of sorbitol and low pH_ex_ seemed to have a synergistic effect as it completely inhibited growth of all strains, including WT. To confirm that these effects were not sorbitol specific, we also tested trehalose as a disaccharide that was shown to be specifically important for membrane stabilisation during salt, oxidative, desiccation and ethanol stress [33,34,35]. Trehalose had absolutely no effect on the growth phenotype of the mutant strains (Figure 3C). We therefore conclude that osmotic stabilisation does not alleviate the growth defect of cell wall and calcium mutants at low pH_ex_, indicating that inorganic acids affect the cell wall differently than cell-wall-biosynthesis-inhibitory compounds such as Congo Red or Calcofluor White. Despite this, the *hog1*Δ mutant is sensitive to low pH_ex_ [25] and many genes responsive to inorganic acid stress are not upregulated in this mutant [27]. It appears that activation of the HOG regulatory pathway is part of this stress tolerance process.

### 3.3. Low pH_ex_ Induces Cell Lysis in CWI Deficient Cells

To assess specific targets in the cell wall, we analysed the proteins released from the cells upon low pH treatment. We cultured wild-type and *slt2*Δ cells for 8 h at pH_ex_ 5.0 and pH_ex_ 2.5 and used mass spectrometry to identify the proteins specifically released upon low pH treatment (Figure 4 and Appendix A). In the *slt2*Δ culture at pH_ex_ 2.5, we identified a large number of cytoplasmic proteins, strongly indicative of extensive cell lysis. Although we do not have truly quantitative data, the number of peptides detected for each protein gives an indication of abundance. Fourteen proteins were identified in all four conditions tested (Figure 4), and they were most abundant in all cases in the wild-type sample at pH 5.0. All are cell surface related proteins: covalently linked cell wall proteins (Ccw14, Ecm33, Ygp1, Zps1 and Pts1), cell wall modifying enzymes (Bgl2, Gas1, Gas5 and Cts1), plasma membrane proteins involved in cell surface stress sensing (Msb2 and Wsc4) and cell surface proteins (Pry1, Bar1 and Pho3). Seventeen more were present in both WT samples, and these too were cell surface proteins. Two had an apparently increased release into the supernatant at pH_ex_ 2.5 (Cwp1p and Yps3p) and six were present at pH_ex_ 2.5 but not at pH_ex_ 5.0 (Hsp12, Tdh3, Trx1, Zeo1p, Sun4 and Plb1). The first three of these are cytoplasmic and might be indicative of cell lysis. On the other hand, Tdh3 peptides were also found to be secreted as stress induced antimicrobial peptides [36]. The other three, as well as Cwp1p and Yps3p, are cell surface modulating or stress sensing cell surface proteins and may play a regulatory role. However, we did not find a specific class of proteins (sulphur-bridge linked, GPI-linked or coupled to β-1,3-glucan through an alkali sensitive O348 glycosylation dependent binding) to be released because of the acid stress. This might be because of (PKC-Slt2p regulated) cell wall remodelling [20]. The experiment with the *slt2*Δ mutant did not resolve this because of the extensive cell lysis and consequent abundance of cytoplasmic protein material.

### 3.4. Effects of Low pH_ex_ on Intracellular Acidification 

Weak organic acids affect cells growth in part because of a reduction of pH_c_ [3]. However, we previously observed that a reduction of pH_ex_ caused by inorganic acids does not influence pH_c_ in WT yeast. To understand the reasons for growth inhibition of the CWI and Ca^2+^ signalling mutants, we simultaneously assessed growth and pH_c_ during low pH_ex_ stress. We transformed wild type and mutants with a plasmid carrying the pH sensitive GFP ratiometric pHluorin [5,37]. As representatives of CWI and Ca^2+^ signalling, we selected the CWI Map kinase Slt2 and the Ca^2+^ channel subunits Mid1 and Cch1, all sensitive to pH_ex_ 2.5 in YNB (Figure 1 and Figure 2). The mutant *pma1-007*, with reduced proton pumping capacity, was also included as a control due to its reduced pHc and accordingly reduced growth rate (Figure 1 and [12]). Both WT and the mutants *mid1*Δ, *cch1*Δ and *pma1-007* showed normal growth profile in medium adjusted to pH_ex_ 5.0 (Figure 5A). Upon inoculation in fresh YNB medium at pH_ex_ 5.0, the initial pH_c_ was approximately 7.2 for most of strains, except *pma1-007*, which, as expected, had a lowered pH_c_ of 7.0 (Figure 5B). Cytosolic pH slowly decreased during growth and rapidly dropped when glucose was depleted until it reached a value of approximately 6.0 (Figure 5B). WT growth was slightly reduced at pH_ex_ 2.5. While *pma1-007* was mildly affected, *mid1*Δ and *cch1*Δ severely affected and *slt2*Δ showed no growth (Figure 5C). Directly upon inoculation, the pH_c_ of the CWI and Ca^2+^ mutants was still similar to that of WT (7.03 ± 0.00), *slt2*Δ (7.07 ± 0.01), *mid1*Δ (7.01 ± 0.01) and *cch1*Δ (7.03 ± 0.01), and the kinetics of pH_c_ variation in WT cells was practically the same in both initial pH_ex_ media (Figure 5D). The initial pH_ex_ of the *pma1-007* mutant was 6.8 at pH_ex_ 2.5 and remained practically unchanged (Figure 5D). Such pH_c_ value was only reached by the WT culture at pH_ex_ 5.0 when it had a low growth rate while approaching stationary phase (Figure 5A,B), which might explain why *pma1-007* mutant grew slowly at pH_ex_ 2.5 (Figure 5C). On the other hand, shortly after inoculation, we observed a loss of fluorescence in the *slt2*Δ, *mid1*Δ and *cch1*Δ mutants, leading to low fluorescence values that no longer reliably represented pH_c_. Visual inspection of the cells revealed an almost complete loss of fluorescence in the *slt2*Δ cells after 2 h at pH_ex_ 2.5 (our unpublished observations).

The loss of fluorescent signal suggested a loss of cell integrity caused by the low pH_ex_. To test whether this coincided with a loss of culture viability, cells were collected after 16 h of incubation at pH_ex_ 2.5 and re-inoculated to fresh medium adjusted to pH_ex_ of 5.0. For WT and *pma1-007*, we readily observed growth (Figure 5E) and stable pH_c_. Ca^2+^ signalling mutants *mid1*Δ and *cch1*Δ resumed growth after a long lag phase (Figure 5E), after which a reliable and normal pH_c_ could again be observed. This suggests that only a fraction of the cells was able to recover. On the other hand, *slt2*Δ never resumed growth (Figure 5E), as previously reported [25]. This indicated that the cellular damage caused by low pH_ex_ is irreversible when CWI mechanism is not operational. We corroborated this interpretation with cell viability counts of cultures first exposed to pH_ex_ 2.5. WT cultures retained full viability during cultivation at pH_ex_ 2.5 for 8 h, whereas the *slt2*Δ mutant rapidly lost viability, with a reduction in survival to 89% in the first 2 h and to about 5% after 8 h (Figure 5F). Together, this suggests that the capacity to remodel the cell wall is essential for maintenance of cellular integrity and thus viability during exposure to low pH_ex_.

### 3.5. Low pH_c_ Protects Cells from Loss of Viability at Extreme Low pH_ex_

While low pH_ex_ did not affect pH_c_ in WT, it strongly affected pH_c_ in the *pma1-007* mutant (Figure 5D), despite the fact that this reduction in pH_c_ by itself was not lethal (Figure 5C,E). However, a reduction of pH_ex_ to 2.5 was lethal for mutants with defects in cell wall repair and, to a lesser extent, Ca^2+^ signalling. To investigate the pH limits of growth in wild type, we cultivated the yeasts at even lower pH_ex_ conditions. We observed a strong reduction of growth in WT at pH_ex_ 2.0 and 1.5 (Figure 6A), which coincided with an immediate decrease of pH_c_ upon inoculation of the media (Figure 6C). The *slt2*Δ mutant showed strongly reduced growth at pH_ex_ 2.0 and 1.5, but, unexpectedly, at pH_ex_ 1.5, we observed a slow but consistent increase in OD600 over 15 h of the time course of the experiment (Figure 6B). At pH_ex_ 1.5, *slt2Δ* retained the fluorescence intensities that could be readily interpreted and pH_c_ that remained stable around a value between 6.7 and 6.8 (Figure 6D). This suggested a long-term cellular integrity at pH_ex_ 1.5, which was absent at pH_ex_ 2.5. Indeed, while none of WT, *slt2*Δ and *pma1-007* showed significant growth at pH_ex_ 1.5 (Figure 6E), all strains retained a fluorescent signal, as well as the ability to resume growth when transferred to medium adjusted to pH_ex_ 5.0 (Figure 6F). In addition, *slt2*Δ cultures retained around 80% of cell viability after 8 h at pH_ex_ 1.5 (Figure 6G), about 15 times higher than the viability shown at pH_ex_ 2.5 (Figure 5F). Thus, the *slt2*Δ mutant is much less affected by pH_ex_ 1.5 than by pH_ex_ 2.5, while the stress dosage is 10 times higher (32 mM vs. 3.16 mM free H^+^ in the medium).

When trying to understand what could be responsible for the increased survival, we noticed that both WT and *slt2*Δ showed a rapid decrease in pH_c_ when transferred to media at pH_ex_ 1.5 (Figure 6C, D). Such a drop by itself does not lead to loss of viability, but has been shown to be associated with a growth rate reduction [12], which in turn can improve stress tolerance [38]. We hypothesised that the decrease in pH_c_ might somehow protect the cells from the external pH stress. To test this hypothesis, we manipulated the intracellular pH of the cells incubated at pH_ex_ 2.5 to compare to the observed at pH_ex_ 1.5. We did so by addition of a small dose of sorbic acid (HS) to the media. In a low pH environment, this organic acid is predominantly protonated and therefore uncharged and can enter the cells by diffusion, which results in a decrease in pH_c_ [3]. For this experiment, WT, *slt2*Δ and *pma1-007* cultures were transferred to pH_ex_ 2.5 and 1.5 with or without addition of HS. As expected, HS in combination with sulphuric acid (pH_ex_ 2.5) reduced pH_c_ of the three strains by about 0.5 pH units (Table 2). At pH_ex_ 1.5, pH_c_ was already reduced 0.3 pH units for all three strains, compared that measured at pH_ex_ 2.5 and addition of HS further reduced pH_c_ 0.6–0.7 pH units (Table 2). The presence of HS did not affect the growth of *slt2*Δ at pH_ex_ of 2.5 or 1.5 (Figure 7A). However, when those cultures were transferred to fresh media at pH_ex_ 5.0, we observed a growth recovery for cultures incubated for 16 h at pH_ex_ 1.5 as well as those incubated at pH_ex_ 2.5 in the presence of HS (Figure 7B). Lastly, and in line with the observe growth recovery, HS also acid enhanced culture viability, as observed by CFU counts: viability of the *slt2*Δ cultures at pH_ex_ 2.5 + HS after 8 h of incubation was nearly 90%, similar to the viability of this mutant at pH_ex_ 1.5 and that of the WT (Figure 7C). Therefore, the shutdown of the cellular processes imposed by the extreme pH_ex_ impaired even more cell growth in such harmful environment, maintained cell longevity even for *slt2*Δ mutant.

### 3.6. Absence of Growth Protects Cells from Low pH_ex_ Lethality

The loss of viability in the absence of a cell wall repair mechanism reminds of the mode of action of β-lactam antibiotics, which interfere with cell wall biogenesis and therefore specifically kill growing cells. A reduction in pH_c_ is known to reduce growth rate and therefore might prevent cell lysis caused by a non-reparable cell wall defect. To test whether indeed the effect of low pH_ex_ was growth-dependent, cells were incubated in media without glucose at pH_ex_ 2.5, conditions under which the cells hardly grew (Figure 8A). In such non-dividing condition, WT as well as *slt2*Δ and *pma1-007* mutants maintained a stable fluorescence with ratios indicating low but constant pH_c_ (Figure 8C). This is remarkable, considering that *slt2*Δ lost its fluorescence when cultivated at pH_ex_ 2.5 (Figure 5C). After 16 h of incubation and transference to fresh media with glucose at pH_ex_ 5.0, all three strains resumed growth (Figure 8B) accompanied by the normal pH_c_ decay upon glucose depletion (Figure 8D). Therefore, in the absence of stimulus for rapid growth (glucose), the deleterious effect of low pH_ex_ is prevented, similar to what we observed at pH_ex_ 1.5 and 2.5 with HS. In conclusion, we showed that, while survival at pH_ex_ 2.5 requires proper cell wall maintenance or repair, a reduction of pH_c_ leading to reduced growth can compensate for the lack of this CWI mediated process.

## 4. Discussion

Changes in extracellular pH arise during yeast growth when the culture naturally acidifies the environment because of metabolism, or in human-generated conditions such as in the process of bioethanol production in which yeast biomass is recycled using sulphuric acid to control the bacterial population [1,2]. Adaptation of yeast to low pH involves several mechanisms including induction of CWI genes [2,22,25] in which action of the PKC pathway is essential [23,24,25]. Here, we worked to establish the cellular effects of inorganic acids and the adaptive responses to tolerate these acids in yeast. All tested strains performed better under low pH conditions in YNB than in YPD, in agreement with complex YPD media containing compounds, such as weak organic acids, that provide an additional stress [5].

It is important to note the differences between the effects of weak organic and inorganic acids in cells. The action of weak organic acids is relatively well known: in a low pH environment, weak organic acids are predominantly protonated, and therefore uncharged, and can enter in the cells by diffusion. The relatively high pH of the cytosol causes a dissociation of the organic acid into the anion and protons, both relatively membrane-impermeable and therefore accumulating inside the cell [6,18]. This causes a reduction of growth because of a reduction of pH_c_ as well as the energy spent to extrude anions and protons to retain homeostasis [21]. For strong inorganic acids, some features should be considered. Initially, because of the impermeability of the plasma membrane to H^+^, predominantly external effects are expected to affect cell wall and plasma membrane. This has been indeed reported by analysis of cell surface by scanning electron microscopy [2]. Later, if the plasma membrane is affected, this might lead to changes membrane potential and/or permeability, allowing influx of H^+^. In this case, we would expect to observe changes in cytoplasmic pH. In line with this model, we observed that, in WT, growth was slightly reduced at pH_ex_ 2.5, but pH_c_ behaviour was quite similar to that at pH_ex_ 5.0 (Figure 5).

The first action of protons at the level of the cell surface fits with the observation that the cell wall integrity PKC pathway is essential for growth at low pH_ex_ (Figure 1). How does the low extracellular pH cause deformations or injuries to the yeast cell surface? We attempted to assess the proteins affected by low pH using mass spectrometry analysis of the proteins released in the medium. While we did identify a major shift in proteins released at pH 2.5 compared to pH 5.0, we could not conclude that a specific sub-set of surface proteins was released (Appendix A), suggesting that it is not a “simple” case of acid hydrolysis of cell wall and its anchoring connections. The fact that the low pH_ex_ lysis could not be remediated with osmostabilisation, which does work for damage induced by the chitin-binding compounds Congo Red and calcofluor white, or for cell wall biosynthetic mutants, suggests that the mechanism is different from known methods of interference with cell wall biogenesis. It is noteworthy that yeast cell had to be actively growing in order for the low pH_ex_ to affect the cell wall, and consequently its viability (Figure 5 and Figure 8). We conclude that it is most likely that the low pH_ex_ interferes with proper biosynthesis of the cell wall during the vegetative cell cycle, analogous to β-lactam antibiotics in bacteria [39]. The cell wall biosynthesis damage could occur for instance at the level of the 1,3-β-glucosyltransferases, enzyme involved in elongation and cross-linking of 1,3-β-glucan to chitin and 1,6-β-glucan and proteins to form the mesh-like structure of the main layer of the cell wall [40,41,42]. As the Gas proteins are anchored through GPI (glycosylphosphatidylinositol) in the outer layer of the yeast plasma membrane, they are exposed to changes of external pH, and the normal vegetative growth isoforms Gas1p and Gas5p have an optimum of activity within pH 3.5–5.0 [43]. We also found hypersensitivity of the *gas1*Δ mutant when it was incubated in media at pH_ex_ 2.5 (Figure 1), indicating that cell wall defects are aggravated when β-glucosyltransferase activity is not sufficient. Therefore, the cells would accumulate lethal injuries in their surface when trying to divide without the supporting mechanism for cell wall organisation. These injuries can turn yeast cells to osmosensitivity, explaining why mutants of the HOG pathway are sensitive to low pHex [25,27]. The interplay between CWI and HOG in responding to CW damages have been described not only for acid stress [27], but also for the biocide polyhexamethylene biguanide (PHMB) [44,45]. Ethanol, also reported to produce damages to CW by affecting its nanomechanical proprieties [46], triggers the HOG pathway that collaborates with PKC/Slt2p pathway for the expression of CWI genes in order to remodelling CW and repair the damages [47]. This connection appears to be made by the action of the Ste11p of the HOG pathway on the Rho1p of the PKC pathway, amplifying the signal for CWI gene expression [48].

Yeast can sense and transduce signals from the cell wall to Rho1p and Pkc1 through five membrane-located sensor proteins Wsc1-3p, Mid2p and Mtl1p [47,49]. Low pH_ex_ affects the cell surface in a way that the damage is most likely sensed by Wsc1p, since of all sensors only the *wsc1*Δ mutant was hypersensitive to low pH_ex_ (Figure 1 and [12]). Similarly, in the filamentous fungus *Aspergillus nidulans*, WscAp and WscBp are required for low pH tolerance [50]. Wsc1p sensor behaves as a linear nanospring that is capable of resisting high mechanical force responding to the cell surface stress [51], but direct effects of altered pH_ex_ on this behaviour have not been tested so far. An acidic environment (pH_ex_ 2.5) leads to mechanical injuries in the yeast cell wall and in the end major changes in cell wall morphology, as evidenced by scanning electron microscope analysis that shows clear deformation of the cell surface [2]. This may lead to conformational changes of Wsc1p sensor that transduce the warning signal downstream in the transduction pathway. This signal then leads to regulatory and transcriptional responses to ensure to remodel the cell wall for maintenance of cellular integrity and thus viability during exposure to low pH_ex_.

In addition to a cell wall damage, we found that Ca^2+^ influx is also affected under inorganic acid stress. The fact that supplementation with Ca^2+^ improved growth of WT specifically at pH_ex_ 2.5 points to a negative effect of low pH_ex_ on the activity of calcium channels, resulting in reduced rate of Ca^2+^ uptake [26]. Such a reduced Ca^2+^ influx could be caused by a reduced activity of the Mid1/Cch1 channel activity, because these channels are voltage-gated and, therefore, sensitive to electrochemical changes in plasma membrane. Indeed, *mid1*Δ and *cch1*Δ mutants were hypersensitive to low pH_ex_ (Figure 2A). Growth of these mutants at pH_ex_ 2.5 was rescued by an addition of Ca^2+^ in the medium (Figure 2C), suggesting that low affinity Ca^2+^ influx systems or the Ecm7 protein might compensate for the defective Ca^2+^ influx in these mutants. An increased calcium concentration in cytoplasm leads to activation of calcineurin, which promotes cell survival under a variety of environmental stress conditions [52] including low pH_ex_ [26]. We found that interfering with calcineurin signalling, either deleting its regulatory subunit *CNB1* or treating wild type yeast with the calcineurin inhibitor FK506 (Figure 2), impaired growth at pH_ex_ 2.5. Together, these results confirmed the previous observation that the Ca^2+^/calmodulin regulated phosphatase calcineurin is relevant to adaptation to low pH_ex_ [26], as it is involved in promoting cell survival to a variety of environmental stress conditions [52]. Once activated, calcineurin dephosphorylates the transcription factor Crz1p that migrates to the nucleus to stimulate the expression of stress response genes. However, since the *crz1*Δ mutant grew as well as the WT on media adjusted to pH_ex_ in the range of 2 to 5 (Figure 2A), the calcineurin pathway seems to act independent of this transcription factor to promote growth under inorganic acid stress. Thus, the downstream target of calcineurin that works for acid tolerance is not known and might be a matter of future investigation. One possibility is that the involvement of Ca^2+^ might be related to its importance for the correct processing and trafficking of proteins in the secretory pathway. In this sense, we found that *pmr1*Δ mutant, which lacks the Golgi membrane P-type ATPase ion pump Pmr1p [53], was hypersensitive to low pH_ex_. Hence, it is reasonable to suppose that the involvement of Ca^2+^ in the yeast response to the injuries caused by low pH_ex_ might be related to its action in protein trafficking.

We established that both Ca^2+^ signalling and CWI signalling are essential for growth at low pH. To understand the interaction between these signalling pathways, we remark the following: Ca^2+^ remediation of growth requires the presence of Slt2p (Figure 2C), suggesting that CWI activation is downstream of the Ca^2+^ signal. In line with this, Slt2p can be activated in a manner depending on transport of Ca^2+^ by Pmr1p into Golgi apparatus [53]. This affects downstream Slt2p signalling, which serves to fortify the cell surface and in turn can activate Cch1p and Mid1p [53,54], thus leading to a positive feedback circuit. The fact that Wsc1p is also required reveals that either Ca^2+^ signals and Pkc1 via Wsc1 are activated in parallel or the Ca^2+^ signal and the mechanical sensor are active at different times, where Ca^2+^ would lead to an initial response in seconds [26] and Wsc1p would be required for the sustained response after minutes [23]. Moreover, when the surface is damaged by the low pH_ex_, it could initiate cell wall adaptation response through the Rho1-activated protein kinase C (Pkc1) for the activation of the CWI genes as well as to intensify the influx of Ca^2+^. This calcium dependent mechanism could be responsible, among other things, for the increment of CWI protein trafficking and their localisation at cell surface to repair the structural damages caused by medium acidification.

The most unexpected observation was that a further reduction of pH_ex_ could rescue the lethality of the *slt2*Δ mutation: when the cells of WT, *slt2*Δ and *pma1-007* mutant were incubated in media at pH_ex_ 1.5, we observed a strongly reduced growth (Figure 6E). The *slt2*Δ mutant could not grow at pH_ex_ 2.5, did not recover growth in a fresh media at pH_ex_ 5 and rapidly lost viability (Figure 5). Unexpectedly, at pH_ex_ 1.5, we observed that the cells retained viability (Figure 6G). Thus, the *slt2*Δ mutant is much less affected by pH_ex_ 1.5 than at pH_ex_ 2.5, while the stress dosage is 10 times higher (approximately 32 mM free H^+^ vs. 3.2 mM free H^+^, respectively). We remarked that both wild type and *slt2*Δ when transferred to media at pH_ex_ 1.5 showed a rapid decrease in pH_c_, which did not occur at pH_ex_ 2.5. Why does a pH_ex_ 1.5 in the cultures lead to a drop in pH_c_ of the cells, if the membranes are impermeable to protons, as we observed during exposure of the yeast cells at pH_ex_ 2.5? We could explain that circumstances looking to the electrochemical proton gradient formed between the outside and the inside of the cells. The main regulator of pH_c_ homeostasis in *S. cerevisiae* yeast is the P2-type H^+^-ATPase Pma1p [55]. Pma1p pumps H^+^ ions out of the cell using ATP hydrolysis at a 1:1 ratio [56], which creates an electrochemical proton gradient that regulates proper cytoplasmic pH and drives the secondary import of nutrients across the plasma membrane [8,57]. pH_c_ is also a critical component of the total electro-chemical gradient, which is responsible for the transport of molecules across membranes [6]. In our experiments, as a control, we used *pma1-007*, a mutant with only 50% of the expression and activity of the proton pumping plasma membrane ATPase Pma1p [29,30]. This mutant cannot maintain proper pH_c_ at low pH_ex_ [12]. We noticed that, among the strains tested in this study, only *pma1-007* showed a drop in initial pH_c_ when the cells were transferred to medium at pH_ex_ 2.5 and even at pH_ex_ 5.0 (Figure 5). This indicates that the cause of the drop is not increased influx of H^+^, but rather reduced efflux. For Pma1p to pump-out protons, the free energy yield of ATP hydrolysis must be at least equal to the free energy cost of proton translocation across the membrane, which is in this case against the PMF. Using the Nernst equation, we can calculate the PMF as:PMF = ΔGv + ΔGc = zFV + 2.303RTΔpH(1)
At equilibrium ΔG_ATP_ + PMF = 0(2)

Taken together, we can calculate the pH gradient against which protons can be pumped using hydrolysis of ATP as:ΔpH = −(ΔG_ATP_+zFV)/2.303RT(3)

Unfortunately, the exact membrane potential in yeast is not known, but reported values are between −50 and −110 mV [58] or even down to −200 mV [38]. With the standard free energy of hydrolysis for pH of −30.5 kJ/mol, we can estimate that 2.6 is the minimal pH_ex_ value that still allows proton pumping with a pH_c_ of 7.0 at a membrane potential of −50 mM, around 3.6 at a membrane potential of −110 mV and around 5.1 at the extreme reported value of −200 mV. While the first value is remarkably close to the growth/no growth transition in our experiments, the other two are much higher than the pH_ex_ at which we still observe growth. When we take into account more physiological ADP, ATP and phosphate concentrations, the free energy of hydrolysis for ATP would be closer to −40 kJ/mol and the minimal pH_ex_ values that would allow ATP hydrolysis-driven proton pumping would be 0.8, 1.8 and 3.2 for membrane potentials of −50, −110 and −200 mV, respectively. The last one does not fit with the growth and pH_c_ homeostatic capacity that we still observe at a pH_ex_ of 2.5, indicating that the membrane potential must be between −5 and −110 mV. Although we cannot make any definitive statements, it might well be that at pH_ex_ values around 2.0 the H^+^ pump Pma1p cannot function because of thermodynamic limitation. Cytoplasmic acidification could then be due to glycolytic activity [6] and also explains the reduction of growth in WT in media at pH_ex_ 2.5 (Figure 2B and [12]).

A reduction of internal pH by itself does not lead to loss of viability, but has been shown to be associated with a growth rate reduction [12], which in turn can improve stress tolerance [59]. We hypothesised that the decrease in pH_c_ might somehow protect the cells from the external pH stress. To test this hypothesis, we manipulated the intracellular pH of the cells incubated at pH_ex_ 2.5 by addition of a small dose of sorbic acid to the media. The addition of sorbic acid to the media caused a drop in pH_c_ of the strains and indeed reduced growth rate and rescued viability of the *slt2*Δ mutant, which reached a similar performance as at pH_ex_ 1.5. A reduction of pH_c_, likely through a reduction in growth rate, therefore enables the cells to survive at low pH_ex_. Indeed, reduction of growth rate caused by glucose deprivation had the same effect. We explain this as follows: First, a reduction in growth rate leads directly to an activation of the environmental or general stress response [59,60] and increased investments in cellular robustness [38] which also allows adaptation to the low pH environment [2,25,61]. Second, because the defects to the cell wall caused by low pH_ex_ become apparent during growth, a reduction of growth rate therefore allows better repair of the wall, even in the absence of the PKC activated CWI.

Our data together lead us to propose a model of action and yeast response to inorganic acids (Figure 9). The low pH directly affects the cell wall that activates PKC/Slt2p pathway to express CWI proteins. Through the combined action of the increased internal concentration of Ca^2+^ on the efficiency of protein translocation to cell surface, yeasts are able to induce cell wall fortification to allows adapted growth. If for some reason the PKC pathway is not activated, cells eventually lyse. Remarkably, a reduction of cytosolic pH prevents this lysis, likely because of a reduction of growth rate and the activation of the general stress response, expanding cell longevity in such extreme environment. Hence, when the environmental conditions return to being less harmful, the yeast cells restart growth. Such pattern can explain the behaviour of saprophytic yeasts exposed to sudden changes in the environment or pathogenic yeasts exposed to cell wall damaging biocidal agents under changing pH conditions. It can also help to improve the resistance of probiotic yeasts during the harmful journey in the gastric compartment. Lastly, it can contribute for better understanding the metabolic adjustment of cancer cells in response to changes in the external pH.

## Figures and Tables

**Figure 1 genes-11-00656-f001:**
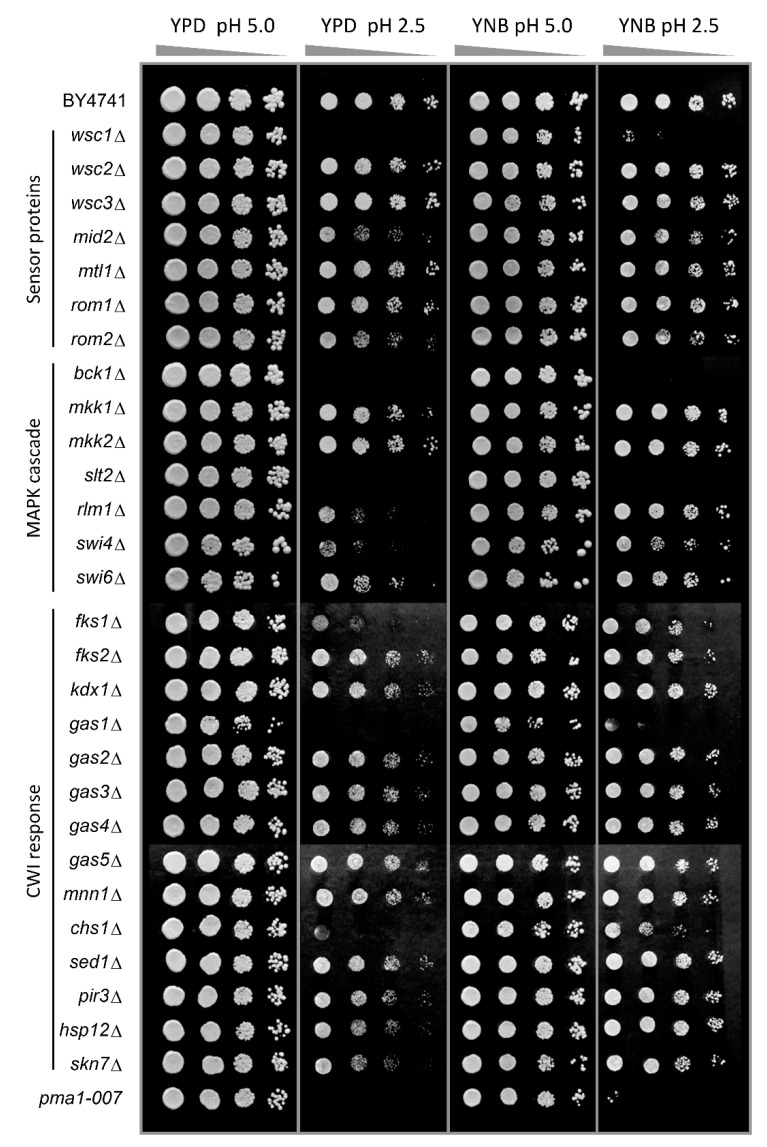
Screening for low pH_ex_ sensitive mutants involved in cell wall signalling and biosynthesis. *S. cerevisiae* BY4741 wild type strain and isogenic mutants were cultivated in YPD medium and serial dilutions (1/10) of yeast cultures were spotted on YPD or YNB agar media adjusted to pH 5.0 or 2.5 with sulphuric acid. Plates were incubated for two days at 30 °C. The mutant *pma1-007* with reduced activity of plasma membrane H^+^-ATPase pump was used as no-growth reference at low pH.

**Figure 2 genes-11-00656-f002:**
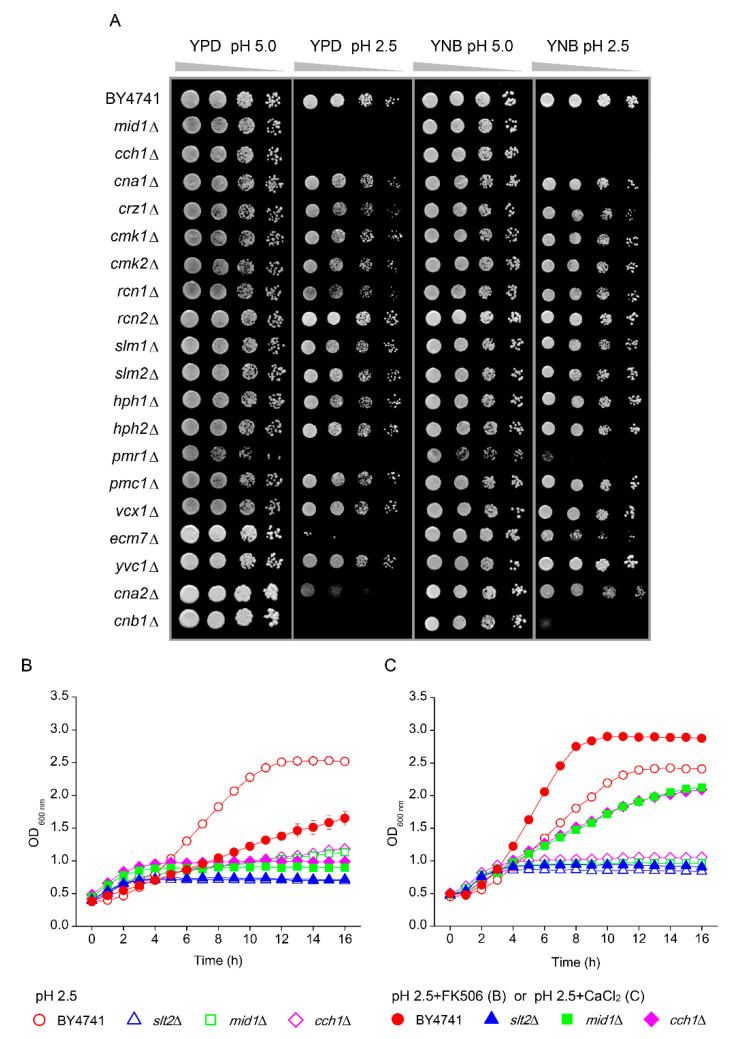
Screening for the involvement of Ca^2+^ transport and signalling pathway on the yeast tolerance to low pH_ex_. (**A**) *S. cerevisiae* BY4741 wild type strain and isogenic mutants were cultivated in YPD medium and serial dilutions (1/10) of yeast cultures were spotted on YPD or YNB agar media adjusted to pH 5.0 or 2.5 with sulphuric acid. Plates were incubated for two days at 30 °C. (**B**) Cells of BY4741 (red circles) and its isogenic mutants *slt2*Δ (blue triangles), *mid1*Δ (green squares) and *cch1*Δ (pink diamonds) were cultivated in YNB adjusted to pH 2.5 with sulphuric acid (open symbols) or pH 2.5 in the presence (closed symbols) of the calcineurin inhibitor FK506 at 2 µL/mL. (**C**) Cells of BY4741 (red circles) and its isogenic mutants *slt2*Δ (blue triangles), *mid1*Δ (green squares) and *cch1*Δ (pink diamonds) were cultivated in YNB adjusted to pH 2.5 with sulphuric acid (open symbols) or pH 2.5 in the excess of Ca^2+^ at 37 mM CaCl_2_ (closed symbols).

**Figure 3 genes-11-00656-f003:**
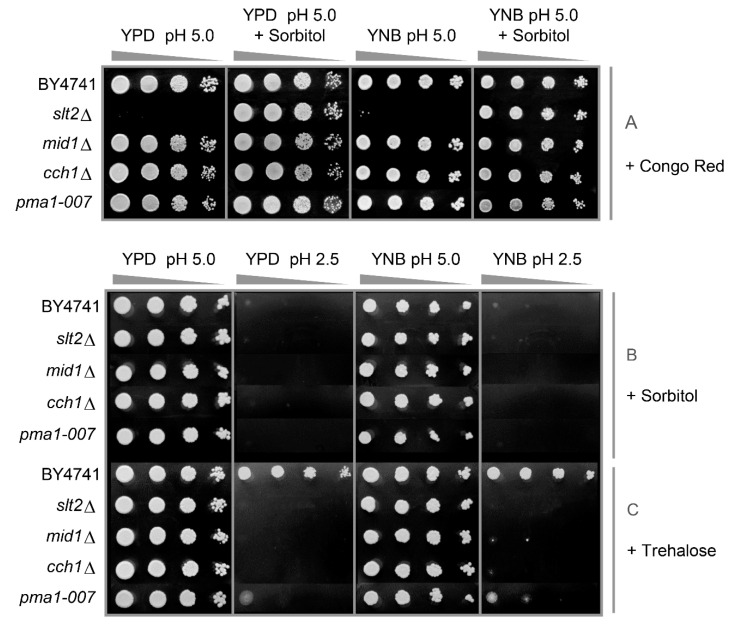
Analysis of the effect of cell wall protective (sorbitol or trehalose) or damaging (Congo red) agents on the tolerance of *S. cerevisiae* to low pH_ex_. Cells of BY4741 and its isogenic mutants *slt2*Δ, *mid1*Δ, *cch1*Δ and *pma1-007* were cultivated in YPD and serial dilutions (1/10) of yeast cultures were spotted on YPD or YNB agar media adjusted to pH 5.0 or 2.5 with sulphuric acid containing Congo red 100 µg/mL: in absence or presence of sorbitol 1 M (**A**); solely containing sorbitol 1 M (**B**); or trehalose 1 M (**C**). Plates were incubated for two days at 30 °C.

**Figure 4 genes-11-00656-f004:**
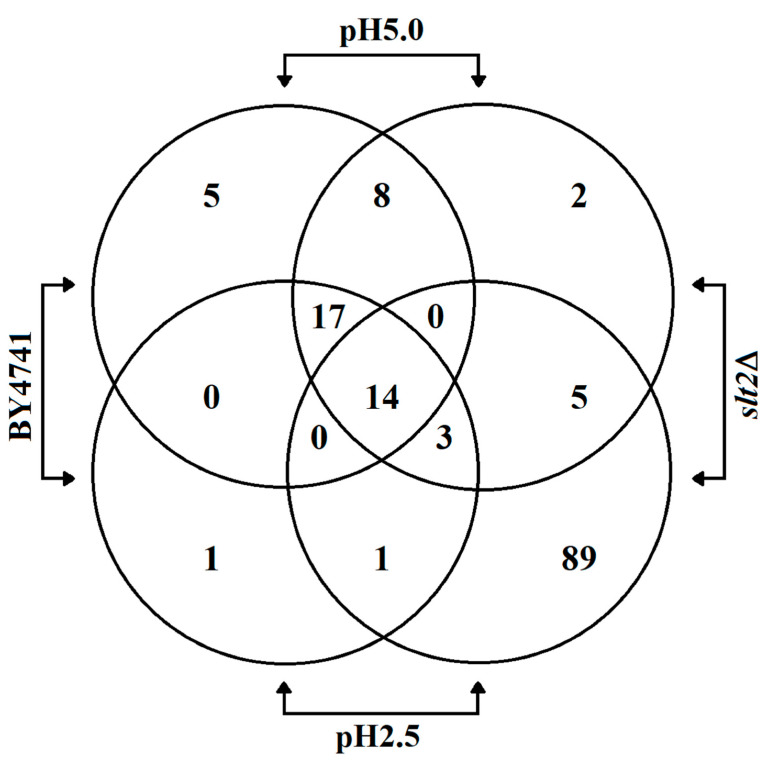
Quantification of overlapping peptides identified in the supernatant of *S. cerevisiae* BY4741 wild type strain and its isogenic mutant *slt2*Δ cultivated in YNB medium adjusted to pH 5.0 or 2.5 with sulphuric acid.

**Figure 5 genes-11-00656-f005:**
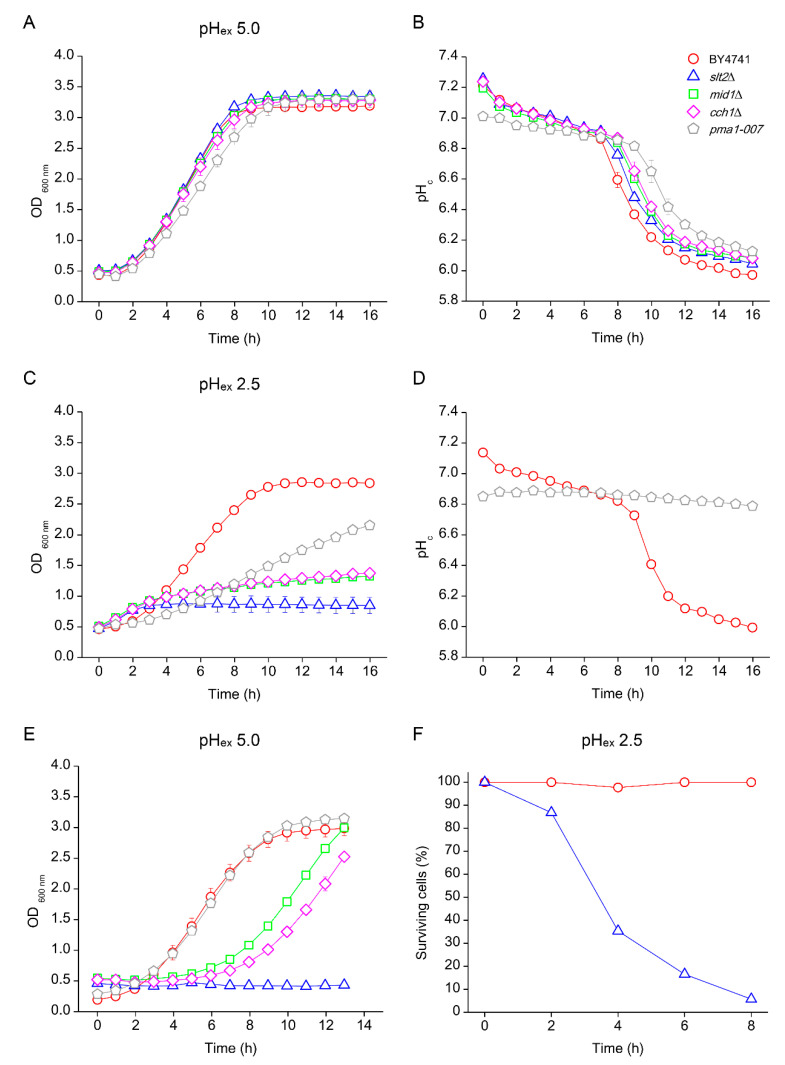
Effect of low external pH (pH_ex_) on growth profile (**A**,**C**) and cytosolic pH (pH_c_) (**B**,**D**) of *S. cerevisiae.* Cells of BY4741 wild type strain (red circles) and its isogenic mutants *slt2*Δ (blue triangles), *mid1*Δ (green squares), *cch1*Δ (pink diamonds) and *pma1-007* (grey pentagon) were cultivated in YNB adjusted to pH 5.0 (**A**,**B**) or pH 2.5 with sulphuric acid (**C**,**D**). Samples were collected from the end of cultivation in YNB pH 2.5 (**C**) and reinoculated to YNB pH 5.0 (**E**) or at different times of cultivation in YNB pH 2.5 and quantified for the percentage of viable cells (**F**).

**Figure 6 genes-11-00656-f006:**
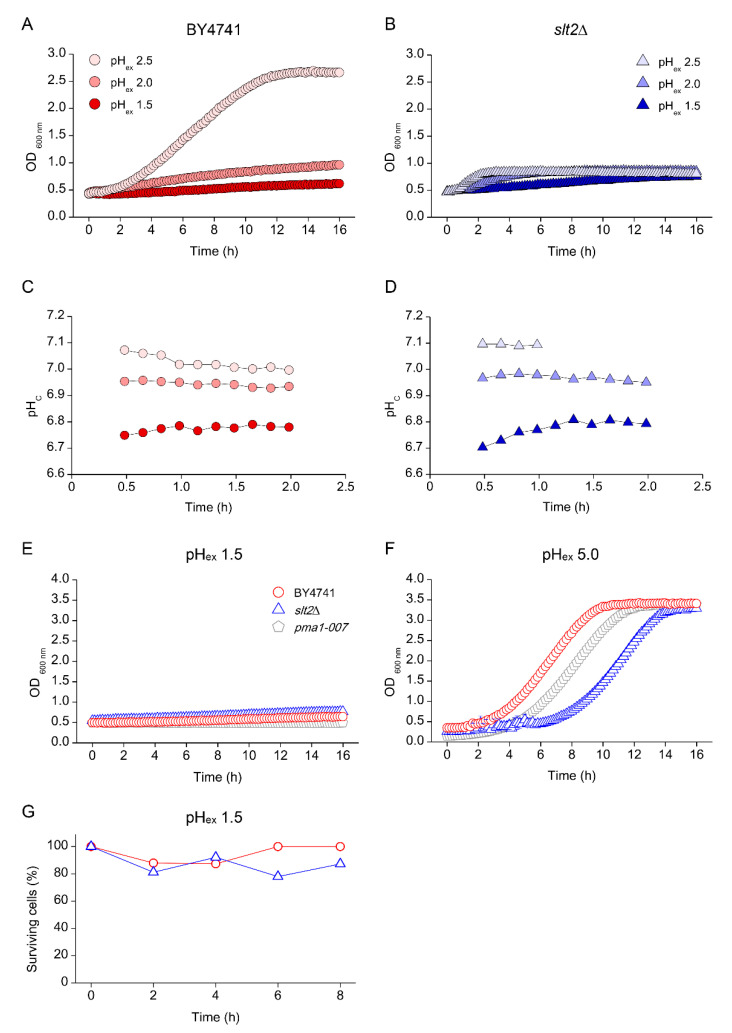
Effect of extreme low external pH (pH_ex_) on growth profile *S. cerevisiae.* Cells of BY4741 wild type strain (red circles) and its isogenic mutants *slt2*Δ (blue triangles) were cultivated in YNB adjusted with sulphuric acid to pH 2.5 (light red or blue), pH 2.0 (medium red or blue) or pH 1.5 (dark red or blue) and growth profile (**A**,**C**) and cytosolic pH (pHc) (**B**,**D**) were defined. Afterwards, BY4741 (red circles) and its isogenic mutants *slt2*Δ (blue triangles) and *pma1-007* (grey pentagon) were cultivated in YNB adjusted to pH 1.5 (**E**) and at the end of cultivation their cells were transferred to YNB pH 5.0 (**F**). Cells were collected at different times of cultivation in YNB pH 1.5 (**E**) and tested for the percentage of viable cells (**G**).

**Figure 7 genes-11-00656-f007:**
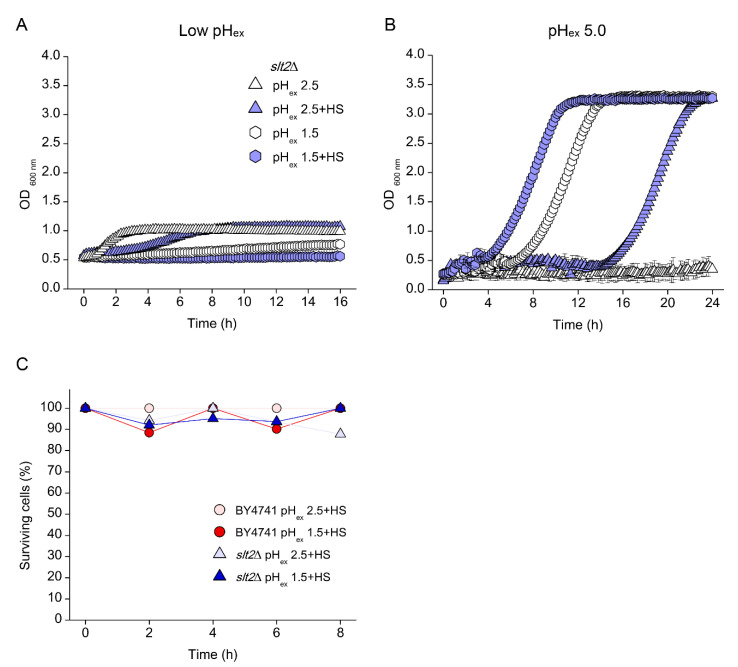
Effect of sorbic acid on the growth profile of *S. cerevisiae slt2*Δ mutant cultivated in low external pH (pH_ex_) medium. (**A**) Cells of were cultivated in YNB adjusted to pH 2.5 (blue triangles) or pH 1.5 (blue hexagon) with sulphuric acid in the absence (open symbols) or presence (closed symbols) of sorbic acid at 1.2 mM. (**B**) At the end of cultivations, their cells were collected and transferred to YNB pH 5.0. (**C**) Cultures of BY4741 (circles) and *slt2*Δ (triangles) strains in YNB containing sorbic acid and adjusted with sulphuric acid to pH 2.5 (light red and blue) or 1.5 (dark red and blue) were evaluated for the percentage of viable cells.

**Figure 8 genes-11-00656-f008:**
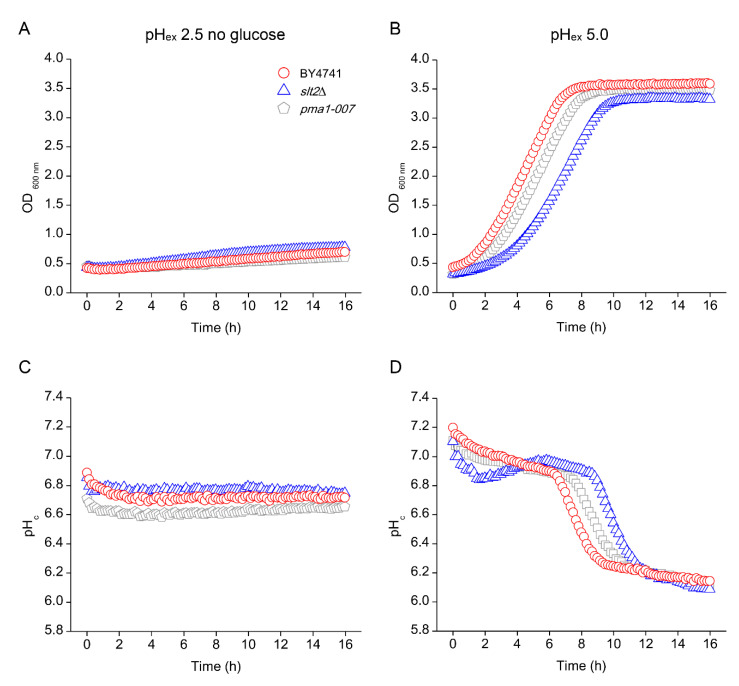
Effect of glucose starvation on the tolerance of *S. cerevisiae* to low external pH (pH_ex_). Cells of *S. cerevisiae* BY4741 wild type strain (red circles) and its isogenic mutants *slt2*Δ (blue triangles) and *pma1-007* (grey pentagon) were cultivated in glucose-lacking YNB adjusted with sulphuric acid to pH 2.5 for measurement of cell growth (**A**) and cytosolic pH (pH_c_) (**C**). At the end of cultivations, cells were collected and transferred to glucose-containing YNB adjusted to pH 5.0 and cell growth (**B**) and pHc variation (**D**) were measured.

**Figure 9 genes-11-00656-f009:**
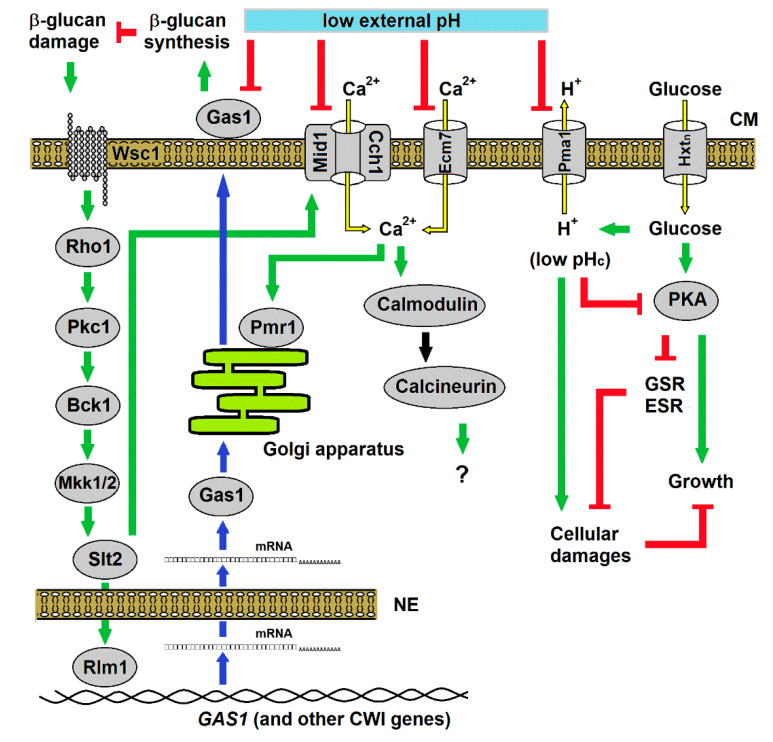
Biological model proposed for the yeast response to stress caused by inorganic acids. Low external pH (pH_ex_) inhibits the action of the β-1,3-glucanosyltransferase Gas1p, resulting in damage to the cell wall β-glucan layer. The damage signal is transduced to PKC pathway through the nanomechanical sensor Wsc1p to activate the cell wall integrity (CWI) genes, including *GAS1*. In parallel, this pathway induces Ca^2+^ uptake by Slt2p-mediated activation of the Mid1p/Cch1p channel, and the imported Ca^2+^ triggers the activation of calcineurin pathway whose function in acid response still unknown. However, Ca^2+^ uptake is counteracted by the inhibitory action low pH_ex_ on Ca^2+^ channels. In consequence, limitation of Ca^2+^ influx jeopardises the activation of the P-type ATPase ion pump Pmr1p responsible for transport of Ca^2+^ and Mn^2+^ to the Golgi apparatus, ions that are necessary for protein processing and trafficking through the secretory pathway, such as those involved in cell wall maintenance and repair. Low pH_ex_ also inhibits the H^+^ efflux pump Pma1p, resulting in the decrease of the cytosolic pH (pH_c_) by the lack of extrusion of H^+^ produced by glucose metabolism. Hence, low pH_c_ reduces the PKA pathway [1,9], causing reduction or impairment in cell growth and release of general stress response (GSR) or environmental stress response (ESR) that work for repairing the cellular damages and maintain cell survival. Symbols refer to flux of molecules (yellow arrows) through transmembrane transporters in the cell membrane (CM), activations (green arrows) or inhibitions (red T-shaped lines) within regulatory pathways, and the genetic flux (blue arrows) from gene transcription, mRNA transport across the nuclear envelope (NE) to the cytosol and protein for translation protein and trafficking inside the Golgi apparatus to outer face of CM.

**Table 1 genes-11-00656-t001:** List of *S. cerevisiae* isogenic mutant strains derived from BY4741 (*MATα his3*Δ*1 leu2*Δ*0 met15*Δ*0 ura3*Δ*0*) used in the present work.

Strain	Biological Function of the Mutated Gene According to *Saccharomyces* Genome Database
*wsc1*Δ	Sensor-transducer of PKC1 MAPKKK pathway involved in maintenance of cell wall integrity (CWI)
*wsc2*Δ	Sensor-transducer of PKC1 MAPKKK pathway involved in maintenance of CWI
*wsc3*Δ	Sensor-transducer of PKC1 MAPKKK pathway involved in maintenance of CWI
*mid2*Δ	O-glycosylated sensor plasma membrane protein acting for CWI signalling
*mtl1*Δ	Putative plasma membrane sensor involved in CWI signalling and stress response
*rom1*Δ	Guanine nucleotide exchange factor (GEF) for Rho1p in the activation of PKC signalling cascade
*rom2*Δ	GEF protein with overlapping function to Rom1p
*bck1*Δ	Mitogen Activating Protein kinase kinase kinase (MAPKKK), first of the PKC signalling cascade
*mkk1*Δ	Target of Bck1p, MAPKK second kinase of the PKC signalling cascade
*mkk2*Δ	MAPKK protein with overlapping function to Mkk1p
*slt2*Δ	Target of Mkk1/2, MAPK third kinase of the PKC signalling cascade
*rlm1*Δ	Target of Slt2p, MADS-box transcription factor involved in the expression of CWI genes.
*swi4*Δ	DNA binding component of the SBF complex (Swi4p-Swi6p) that regulates G1/S checkpoint genes
*swi6*Δ	Transcription cofactor of SBF complex (Swi4p-Swi6p)
*fks1*Δ	Catalytic subunit of 1,3-β-D-glucan synthase involved in CWI mechanism
*fks2*Δ	Catalytic subunit of 1,3-β-D-glucan synthase involved in spore wall biosynthesis.
*kdx1*Δ	Protein kinase implicated in Slt2p signalling pathway of CWI mechanism and mating
*gas1*Δ	Glycosylphosphatidylinositol (GPI)-anchored β-1,3-glucanosyltransferase required for CW assembly
*gas2*Δ	GPI-anchored β-1,3-glucanosyltransferase involved in spore wall assembly
*gas3*Δ	Putative 1,3-β-glucanosyltransferase, member of GAS family
*gas4*Δ	1,3-β-glucanosyltransferase involved with Gas2p in spore wall assembly
*gas5*Δ	1,3-β-glucanosyltransferase; has similarity to Gas1p; localises to the cell wall.
*mnn1*Δ	α-1,3-mannosyltransferase of the Golgi complex required for N-mannosylation of secreted proteins
*chs1*Δ	Chitin synthase I requires that catalysis the transfer of N-acetylglucosamine (GlcNAc) to chitin.
*sed1*Δ	Major stress-induced structural GPI-cell wall glycoprotein associated with translating ribosomes
*pir3*Δ	O-glycosylated covalently-bound cell wall protein required for cell wall stability
*hsp12*Δ	Plasma membrane protein involved in maintaining membrane organisation during stress conditions
*skn7*Δ	Nuclear response regulator and transcription factor that interacts with the Tup1-Cyc8 complex
*pma1-007*	Major plasma membrane H^+^-ATPase pump
*mid1*Δ	Subunit of the Voltage-gated high-affinity calcium Mid1/Cch1 channel
*cch1*Δ	Subunit of the Voltage-gated high-affinity calcium Mid1/Cch1 channel
*cna1*Δ	Catalytic subunit of the Ca^2+^/calmodulin-regulated protein phosphatase calcineurin A complex
*cna2*Δ	Catalytic subunit of the Ca^2+^/calmodulin-regulated protein phosphatase calcineurin A complex
*cnb1*Δ	Calcineurin B; regulatory subunit of calcineurin A complex
*crz1*Δ	Transcription factor regulated by Ca^2+^/calmodulin in response to stress condition
*cmk1*Δ	Calmodulin-dependent protein kinase acting on stress response
*cmk2*Δ	Calmodulin-dependent protein kinase with overlapping function to Cmk1
*rcn1*Δ	Protein involved in calcineurin regulation during calcium signalling
*rcn2*Δ	Protein of unknown function, paralogous to Rcn1p
*slm1*Δ	Phosphoinositide PI4,5P(2) binding protein that acts on cytoskeleton organisation during stress
*slm2*Δ	Phosphoinositide PI4,5P(2) binding protein that forms a complex with Slm1p
*hph1*Δ	Calcineurin substrate tail-anchored ER membrane protein of unknown function
*hph2*Δ	Tail-anchored ER membrane protein of unknown function involved in growth in osmotic and CW stress
*pmr1*Δ	High affinity Ca^2+^/Mn^2+^ P-type ATPase involved in Ca^2+^-dependent protein sorting in the Golgi complex
*pmc1*Δ	Vacuolar Ca^2+^ ATPase involved in depleting cytosolic Ca^2+^ and preventing calcineurin activation
*vcx1*Δ	Vacuolar membrane antiporter with Ca^2+^/H^+^ and K^+^/H^+^ exchange activity for cell ion homeostasis
*ecm7*Δ	Putative integral membrane protein with a role in calcium uptake
*yvc1*Δ	Vacuolar cation channel that mediates vacuolar Ca^2+^ release in response to hyperosmotic shock

**Table 2 genes-11-00656-t002:** Effects sorbic acid (HS) addition in LOFLO medium adjusted to 2.5 or 1.5 (pH_ex_) with sulphuric acid on the cytosolic pH (pH_c_) of the *S. cerevisiae* wild type strain BY4741 and its isogenic mutants *slt2*Δ and *pma1-007*. Numbers represent average ± standard deviation of three biological replicates.

Strains		pH_ex_ 2.5			pH_ex_ 1.5	
		−HS	+HS		−HS	+HS
BY4741		6.96 ± 0.00	6.40 ± 0.00		6.68 ± 0.01	6.10 ± 0.01
*slt2*Δ		7.00 ± 0.02	6.45 ± 0.02		6.70 ± 0.01	6.01 ± 0.01
*pma1-007*		6.83 ± 0.01	6.33 ± 0.02		6.49 ± 0.01	5.90 ± 0.01

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
