# Peer review of "Extreme Low Cytosolic pH Is a Signal for Cell Survival in Acid Stressed Yeast"

_genes, 2020, doi:10.3390/genes11060656_

Round 1
Reviewer 1 Report
This paper describes an interesting and comprehensive study investigating the effect of the low extracellular pH on yeast, regarding intracellular pH and adaptation to growth and survival. The large amount of data presented in this study was translated into a proposal of a biological model for the yeast response to stress caused by inorganic acids.
The study is well-designed, the results are clearly presented and support the conclusions.
Minor suggestions:
- There are a large number of mutants used. A table describing the mutants used in the work with the correspondent description the mutated gene would facilitate following the results (could be inserted in the methods sections).
- Line 489: “Low pHex, or the effect it has on the cell surface.“ Rewrite the sentence, it does not make sense.
Author Response
Minor suggestions:
1) There are a large number of mutants used. A table describing the mutants used in the work with the correspondent description the mutated gene would facilitate following the results (could be inserted in the methods sections).
Response: Added as table 1 (line 80), and the original table 1 was changed to table 2 (legend and references in the text)
2) Line 489: “Low pHex, or the effect it has on the cell surface.“ Rewrite the sentence, it does not make sense.
Response: The sentence " Low pHex, or the effect it has on the cell surface . This damage is most likely sensed by Wsc1p..." was a typing error that was replaced by "Low pHex affects the cell surface in a way that the damage is most likely sensed by Wsc1p..."
Reviewer 2 Report
This manuscript provided a new information about cell stress in yeast and was well-written. However, statistical analysis was not found in this manuscript. Please show significant differences of results. Statistical analysis should be described.
In addition, a recent paper, Udom et al. (2019. Appl Environ Microbiol. 2019;85(15): e00551-19. doi:10.1128/AEM.00551-19) showed HOG pathway plays a collaborative role with the CWI pathway in inducing cell wall remodeling under ethanol stress. Please add this reference to Discussion section and update your discussion. The following paper is also a good review to compare new findings of this manuscript to other papers in this field.
Sanz et al. 2017. The CWI Pathway: Regulation of the Transcriptional Adaptive Response to Cell Wall Stress in Yeast. J Fungi (Basel). 2017;4(1):1. doi:10.3390/jof4010001
Minor comments,
Line 52: Is the sentence “depends on of both” correct ?
Line 72: Please change “environment” to “environmental”.
Line 82: Please show the company and detail of drop-out amino acid supplement powder mixture.
Line 98: Please add rpm of agitation.
Author Response
Comments and Suggestions for Authors
1) This manuscript provided a new information about cell stress in yeast and was well-written. However, statistical analysis was not found in this manuscript. Please show significant differences of results. Statistical analysis should be described.
Response: For the data reported in graphics (Figures 2bc and 5-8), experiments were performed in as three independent biological replicates and standard bars were included. However, for most of the results the bars were so small that became smaller than the symbols. You can see examples of SD bars in Figures 2b and 5ce. The biological phenomena we are relating regarded to differences in growth profile or internal pH variation. In this kind of experiments the shape of the curves matters and our analysis were very focused by comparing growth from no growth, and sometimes conditions in which we had recovered from no growth to growth. In this case, simple SD analysis are confirmatory of the phenomenon. The only case where statistic is worth is in table 1 (which will be new table 2) with the effect of HS in the pHc values. Despite SD had been showed, we can observe a constant drop of ~0.3 units in pHc for all three strains between pHex 2.5 and 1.5 and a constant drop of ~0.5 units in pHc by the presence of HS for all three strains in both pHex. Hence, the biological phenomenon is also clear and reproductive. We think these facts show the reliability and reproducibility of the data.
2) In addition, a recent paper, Udom et al. (2019. Appl Environ Microbiol. 2019;85(15): e00551-19. doi:10.1128/AEM.00551-19) showed HOG pathway plays a collaborative role with the CWI pathway in inducing cell wall remodeling under ethanol stress. Please add this reference to Discussion section and update your discussion. The following paper is also a good review to compare new findings of this manuscript to other papers in this field.
Sanz et al. 2017. The CWI Pathway: Regulation of the Transcriptional Adaptive Response to Cell Wall Stress in Yeast. J Fungi (Basel). 2017;4(1):1. doi:10.3390/jof4010001
Response: To accomodate the suggestion for new references, we added a text from line 489 for contextualisation of the interplay between CWI and HOG suggested by the reviewer: "These injuries can turn yeast cells to osmosensitivity, explaining why mutants of the HOG pathway are sensitive to low pHex [25,27]. The interplay between CWI and HOG in responding to CW damages have been described not only for acid stress [27], but also for the biocide polyhexamethylene biguanide (PHMB) [Elsztein et al 2011; Queiroz et al 2020]. Ethanol, also pointed to produce damages to CW by affecting its nanomechanical properties [Schiavone et al 2016], trigger the HOG pathway that collaborates with PKC/Slt2p pathway for the expression of CWI genes in order to remodelling CW and repair the damages [Udom et al 2019]. This connection appears to be made by the action of the Ste11p of the HOG pathway on the Rho1p of the PKC pathway, amplifying the signal for CWI gene expression [Sanz et al 2017]". Reference order and numbering were modified.
3) Line 52: Is the sentence “depends on of both” correct ? Line 72: Please change “environment” to “environmental”.
Response: both corrected
4) Line 82: Please show the company and detail of drop-out amino acid supplement powder mixture. Line 98: Please add rpm of agitation.
Response: both information provided
Reviewer 3 Report
The manuscript entitled “Extreme low cytosolic pH is a signal for cell survival 2 in acid stressed yeast” presents high-quality work, with a good methodological design and performance. It is proficient written and detailed. Thus, I consider this manuscript need minor changes. My only suggestion is to try to shorten the whole length of the manuscript, especially results and discussion. Also, the format of the figures should be constant through the manuscript (same size font and figure size).
Finally, some graphs in Figures 5 and 6 can be combined in one (figure 5, A and B combined and figure 6, A and B, C and D combined) to make it shorter. It can be considered to move some of these graphs to supplementary material to make the manuscript lighter and easier to read
Author Response
1) My only suggestion is to try to shorten the whole length of the manuscript, especially results and discussion
Response: the manuscript has 24 pages, which would result in eight pages of printed paper. It is quite the average of papers of this type. Without pointing out what should be withdrawn from the text it will be very difficult to shorten the manuscript without stepping out relevant information for the understanding. As it was a kind suggestion from the reviewer, not pointed by the other two, we kindly ask for the maintenance of the text as it is (considering the modifications required).
2) Also, the format of the figures should be constant through the manuscript (same size font and figure size).
Response: figures were prepared with exactly the same format for spot tests on plates and for graphics (type and length of letters and symbols). Differences should be related to numbers of panels, which increase or decrease the whole figure. The resolution was also the same (600 dpi), which allows adjustment by the editorial office.
3) Finally, some graphs in Figures 5 and 6 can be combined in one (figure 5, A and B combined and figure 6, A and B, C and D combined) to make it shorter.
Response: figure 5 has six panels while figure six has seven panels. Merging them will produce a figure with thirteen panels, with numbers and symbols much more smaller than they are now. Besides, they refer to different aspects of the work.
4) It can be considered to move some of these graphs to supplementary material to make the manuscript lighter and easier to read
Response: we understand the concerns of the reviewer with the final length of the paper, but we consider all figures absolutely relevant for the complete understanding of the work since they represent a line of rationale. Figures in supplementary material would indicate that they are only additional or confirmatory results that, normally, are not essential for the core of the work. Hence, we respectfully ask for maintenance of all of them.
Round 2
Reviewer 2 Report
This manuscript is well-improved.
Author Response
1. Lines 54-57 of the introduction argue about the role of Pdr12 in export of anions, however, this should not be generalized because this pump is not involved in extrusion of all WOAs anions (e.g. it is not required for acetate export). Some rephrasing in order to render this aspect clearer would benefit.
RESPONSE: the sentence "The acidification is countered by the activity of the already expressed Pma1p, helped by the induced expression of the ABC transporter Pdr12p to pump out the anions [18,19]" was replaced by "The acidification is countered by the activity of the already expressed Pma1p, helped by the induced expression of the ABC transporter Pdr12p to pump out the sorbate and some moderately lipophilic carboxylic anions [18]. However, this transporter does not act against dicarboxylic and highly lipophilic long-chain fatty acids [19]."
2. Phrasing in Lines 60-63 appear to convey the idea that there is an important role of cell wall damaging response pathways to strong inorganic acids that is not important for response to WOAs (the authors suggest that the mechanisms are different). Although I can understand the point the authors want to convey it is also true that several studies have shown that different WOAs affect the cell wall structure and cause activation of corresponding signalling pathways including of Slt2. Reduced activity of Fks by acetic acid stress has also been shown. Activation of Msn2/Msn4p-regulons, which control GSR, have also been well demonstrated to occur in response to WOAs. Appropriate contextualization is therefore required in order not to convey the idea that cell wall damaging and corresponding adaptive response are a specific hallmark of inorganic acids which is not the case.
RESPONSE: we understand the analysis by the reviewer. However, in other to avoid the deviation from the focus by adding more information about the action of WOA, we think that it is better to remove the comparison between IA and WOA. Therefore, the sentence "For strong inorganic acids such as hydrochloric or sulphuric acids, adaptation of yeast to low pH involves mechanisms different from those acting on the effects of WOA..." was replaced by "For strong inorganic acids such as hydrochloric or sulphuric acids, adaptation of yeast to low pH involves mechanisms that include induction of Cell Wall Integrity (CWI) genes [2,22]..."
3. Line 438 has a typo related with referencing.
RESPONSE: reference removed